# Telling Peer Direct Effects from Indirect Effects in Observational Network Data

**Xiaojing Du** [1]   **Jiuyong Li** [1]   **Debo Cheng** [1]   **Lin Liu** [1]   **Wentao Gao** [1]   **Xiongren Chen** [1]   **Ziqi Xu** [2]

## Abstract

Estimating causal effects is crucial for decision-makers in many applications, but it is particularly challenging with observational network data due to peer interactions. Some algorithms have been proposed to estimate causal effects involving network data, particularly peer effects, but they often fail to tell apart diverse peer effects. To address this issue, we propose a general setting which considers both peer direct effects and peer indirect effects, and the effect of an individual's own treatment, and provide the identification conditions of these causal effects. To differentiate these effects, we leverage causal mediation analysis and tailor it specifically for network data. Furthermore, given the inherent challenges of accurately estimating effects in networked environments, we propose to incorporate attention mechanisms to capture the varying influences of different neighbors and to explore high-order neighbor effects using multi-layer graph neural networks (GNNs). Additionally, we employ the Hilbert-Schmidt Independence Criterion (HSIC) to further enhance the model's robustness and accuracy. Extensive experiments on two semi-synthetic datasets derived from real-world networks and on a dataset from a recommendation system confirm the effectiveness of our approach. Our findings have the potential to improve intervention strategies in networked systems, particularly in social networks and public health.

## 1. Introduction

Causal effect estimation is an important area of study, with the focus on determining cause-and-effect relationships between variables (Imbens & Rubin, 2010; Pearl, 2009). When

[1]University of South Australia, Adelaide, Australia [2]School of Computing Technologies, RMIT University, Melbourne, Australia. Correspondence to: Jiuyong Li <jiuyong.li@unisa.edu.au>, Debo Cheng <debo.cheng@unisa.edu.au>.

*Proceedings of the $42^{nd}$ International Conference on Machine Learning*, Vancouver, Canada. PMLR 267, 2025. Copyright 2025 by the author(s).

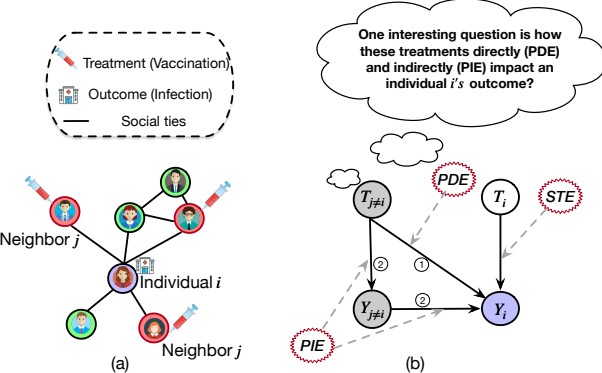

*Figure 1.* (a) An example showing the relationship between an individual $i$ and their neighbors in a network setting about vaccination. (b) A causal graph representing PDE, PIE, and STE. In the diagram, $T_i$ and $Y_i$ indicate the treatment (e.g., vaccination status) and outcome (e.g., infection condition), respectively. The subscript $i$ denotes an individual and $j \neq i$ denotes $i$'s neighbors.

data is collected from sources such as social networks, communication networks, or biological networks, accurately estimating causal effects becomes challenging due to the interconnected nature of individuals (e.g., units, nodes). In a network setting, an individual's outcomes are influenced not only by their own treatment but also by the treatments and outcomes of their neighbors (Sinclair et al., 2012). In network data, as shown in Fig.1, there are different types of causal effects involved with respect to an outcome of interest, including individual outcomes influenced by their own treatment (*self-treatment effects, STE*), and peer effects (arising from the treatments and outcomes of their neighbors), which consist of *peer direct effects (PDE)* and *peer indirect effects (PIE)*. It is important to distinguish and estimate these different types of causal effects in network data, particularly the PDE and PIE.

We use an example from epidemiology (VanderWeele et al., 2012) to show the importance of differentiating between PDE and PIE. Suppose that an individual $i$ is unvaccinated against a disease but can benefit from their peers' vaccination in two ways. First, vaccination of $i$'s peers reduces the likelihood of them transmitting the disease to others (PDE), thereby lowering $i$'s risk of contracting the disease. Second, vaccinated peers are less likely to become infected,

which reduces $i$'s exposure to the disease (PIE). Thus, in the context of vaccination, it is essential to understand how vaccines directly reduce transmission (PDE) and how the effect of vaccination of peers decreases the contagion risk of an individual (PIE). To illustrate this point, consider an extreme case: suppose the vaccination of $i$'s peers does not reduce the likelihood of spreading a disease (PDE is zero). However, if the vaccine reduces the potential of infection, hence reducing $i$'s exposure to the disease, then the PIE is non-zero. In this case, under limited resources, it is advisable to prioritize vaccinating individuals with a higher probability of infection to achieve herd immunity.

A similar example can be found in product promotion campaigns. Let's assume that an individual $i$ has not exposed to an advertisement for a product, but $i$'s peers have. $i$ might decide to purchase the product because their peers received the advertisement, even if the peers did not buy it (PDE); or $i$ might be influenced by the fact that their peers bought the product (PIE). Drawing an analogy to the vaccination example, if an advertisement is "infectious" (i.e., has high PDE), it spreads quickly through the network, reaching untargeted individuals who purchase the product based on their peers' exposure. If an advertisement is "contagious" (i.e., has high PIE), individuals' purchasing behavior is influenced by their peers' purchases of the product. This distinction helps advertisers better understand the effectiveness of their campaigns and allows for more strategic planning of future advertisements.

The existing works do not effectively distinguish between PDE and PIE in network data. A summary of works in peer effect analysis of network data is shown in Table 1, where we see that the works by (Jiang & Sun, 2022; Cai et al., 2023; Chen et al., 2024) do not disentangle PDE and PIE. However, in many cases, particularly in infectious diseases and marketing, it is not enough to simply know that overall peer effects. It is important to distinguish between PDE and PIE, as each has different practical implications. (VanderWeele et al., 2012; Chiba, 2013; Shpitser et al., 2017; Cai et al., 2021) separately consider PDE and PIE, but they primarily focus on two-person transmission scenarios (i.e., one-to-one interactions), with the first two studies assuming binary outcomes. The one-to-one interaction restricts their applicability to real-world networks where multi-to one relationships are prevalent. The details of the limitations of these methods are discussed in the Related Work section.

To address these limitations, we consider a more general setting for observational network data, estimating the PDE and PIE at the group level, and additionally evaluating the STE within networks, where a group is defined as a set of individuals connected through social ties, rather than treating each individual in isolation. We apply the principles of mediation analysis (Pearl, 2014) and the backdoor crite-

*Table 1.* A comparison of problem settings across different methods.

| Method | PDE | PIE | STE | Relationship Type |
|---|---|---|---|---|
| (VanderWeele et al., 2012) | ✓ | ✓ | ✗ | One-to-One |
| (Chiba, 2013) | ✓ | ✓ | ✗ | One-to-One |
| (Cai et al., 2021) | ✓ | ✓ | ✗ | One-to-One |
| (Shpitser et al., 2017) | ✓ | ✓ | ✗ | One-to-One |
| (Jiang & Sun, 2022) | ✗ | ✗ | ✓ | Many-to-One |
| (Cai et al., 2023) | ✗ | ✗ | ✓ | Many-to-One |
| (Chen et al., 2024) | ✗ | ✗ | ✓ | Many-to-One |
| gDIS (**Ours**) | ✓ | ✓ | ✓ | Many-to-One |

rion (Pearl, 2009) to distinguish these effects, and provide their identification conditions to support the soundness of the estimation. Based on the theoretical analysis, we develop a new algorithm, gDIS, for **g**roup-level P**D**E and P**I**E, PE and **S**TE estimation with network data. To accurately capture the complexity of network effects, gDIS employs a multi-layer GNN to focus on high-order neighbor interference and leverages attention mechanisms (Niu et al., 2021) to account for the varying influence weights of different neighbors on each individual. Furthermore, to fully utilize the structural information of graphs, we integrate the Hilbert-Schmidt Independence Criterion (HSIC) (Ahmad et al., 2021) into the GNN, enhancing the model's robustness and accuracy.

The main contributions of this paper are summarized as follows:

- We propose a framework to distinguish peer effects in observational network data into direct and indirect components at the group level, and provide theoretical analyses of the identification conditions of the causal effects.

- We propose a novel algorithm gDIS to estimate effects in network data by utilizing GNNs and attention mechanisms, enabling a more nuanced and robust analysis of network data.

- We validate the effectiveness and robustness of gDIS on two semi-synthetic datasets based on real-world networks, as well as on recommendation system data, demonstrating the ability of gDIS to accurately estimate different types of causal effects in complex network data.

## 2. Related Work

In this section, we review methods for estimating causal effects in network data that are related to our gDIS method. The methods for peer effect estimation in network data fall in two broad categories: methods which do not differentiate

between direct and indirect peer effects, and methods which do.

**Methods Which Do Not Differentiate between Direct and Indirect Peer Effects.** Many methods have been developed for estimating peer effects without differentiating between direct and indirect peer effects. For example, Forastiere et al. (2021) addressed this issue with a covariate adjustment method using a generalized propensity score (PS) (Feng et al., 2012) to balance both individual and neighborhood covariates. Jiang & Sun (2022) introduced NetEst, a framework that employs graph neural networks (GNNs) to capture feature representations of both individual nodes and their first-order neighbors. Cai et al. (2023) expanded on this by deriving generalization bounds and proposing a joint propensity score approach combined with representation learning via weighted regression. Chen et al. (2024) integrated the targeted learning (Van der Laan & Rose, 2011) into neural network training, developing a causal effect estimator. Ma et al. (2022) developed HyperSCI, which leverages hypergraph neural networks (HGNNs) (Feng et al., 2019) to model interference using a multilayer perceptron (MLP) (Popescu et al., 2009) and hypergraph convolution. However, these methods do not differentiate between the various types of peer effects. It is crucial to differentiate between PDE and PIE, as these two components carry distinct practical implications and inform different strategies for intervention and decision-making.

**Methods differentiating between direct and indirect peer effects.** This line of research focuses on estimating direct and indirect peer effects separately in network data. VanderWeele et al. (2012) and Chiba (2013) used logistic regression and inverse probability weighting (IPW) methods, respectively, to study the effects of vaccination in small family units, such as the impact of a child's vaccination on the mother. Shpitser et al. (2017) developed a chain graph-based method to decompose peer effects into unit-specific components. Cai et al. (2021) evaluated contagion, susceptibility, and infectiousness effects in symmetric partnerships under infectious disease settings. The work presented in (VanderWeele et al., 2012; Chiba, 2013; Shpitser et al., 2017; Cai et al., 2021) primarily focuses on scenarios with only two individuals in the transmission unit, limiting its practical applicability. Additionally, the studies in (VanderWeele et al., 2012) and (Chiba, 2013) focus on binary outcomes only. In contrast, our work focuses on estimating both direct and indirect peer effects at the group level, making it more applicable to real-world scenarios.

## 3. Preliminaries and Background

This section outlines the notations, definitions, problem settings, and assumptions used in the paper. More details of background and notations are provided in Appendix A and

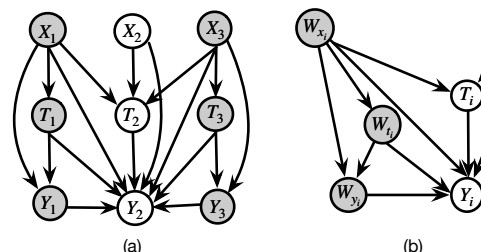

*Figure 2.* (a) An illustration of the causal relationships considered in our work. Node 2 has nodes 1 and 3 as neighbours in the network. The features, treatment, and outcome of node $i$ are represented by $X_i$, $T_i$, and $Y_i$, respectively. (b) The summary causal graph where $W_{x_i}$, $W_{t_i}$, and $W_{y_i}$ represent the aggregated features, treatments, and outcomes of node $i$'s neighbors.

in Table 3 of Appendix B.

### 3.1. Notations

Uppercase letters (e.g., $W_{t_i}$) denote variables, and lowercase letters (e.g., $w_{t_i}$) their values. Bold uppercase letters (e.g., $\mathbf{W}_x$) represent sets, vectors, or matrices, while bold lowercase letters (e.g., $\mathbf{w}_x$) represent their values.

As illustrated in Fig. 2(a), a **network** consists of a collection of nodes (or entities) and edges (or connections) that represent relationships between them. For example, in a social network, individuals serve as nodes, while their friendships or interactions form the edges connecting them. Two nodes are **neighbors** if they are connected by an edge. For simplicity, node indices (e.g., $i$) are used to represent nodes when the context is clear.

The causal relationships in this study are represented by the directed edges between the nodes (variables) in a **causal directed acyclic graph (DAG)**, as shown in Fig. 2(b). The node set is defined as $\mathbf{V} = \mathbf{X} \cup \mathbf{T} \cup \mathbf{Y}$, where $\mathbf{X} = \{\mathbf{X}_1, \ldots, \mathbf{X}_m\}$ represents features, $\mathbf{T} = \{T_1, \ldots, T_m\}$ denotes binary treatments (e.g., vaccination), and $\mathbf{Y} = \{Y_1, \ldots, Y_m\}$ represents continuous outcomes (e.g., immunity levels). Each node $i$ has a feature set $\mathbf{X}_i = \{X_{i1}, \ldots, X_{ik}\}$, where $i \in \{1, \ldots, m\}$.

### 3.2. Definitions of Causal Effects

We focus on estimating Peer Direct Effects (PDE), Peer Indirect Effects (PIE), and Self-Treatment Effects (STE) from network data, defined within the potential outcomes framework (Imbens & Rubin, 2010).

In the potential outcomes framework, $Y_i(t')$ is the potential outcome when treatment $T$ takes the value $t'$; when there are two treatments, $Y_i(t'_1, t'_2)$ denotes the potential outcome under $T_1 = t'_1$ and $T_2 = t'_2$. The individual treatment effect (ITE) for unit $i$ is defined as $\text{ITE} = Y_i(t') - Y_i(t)$, and the

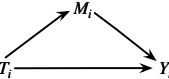

*Figure 3.* A classic causal mediation graph consists of three variables: $T_i$ (treatment), $M_i$ (mediator), and $Y_i$ (outcome).

average treatment effect (ATE) is ATE $= \mathbb{E}[\,Y_i(t') - Y_i(t)\,]$, where $t'$ represents the intervention (treatment) state and $t$ represents the no-intervention (control) state.

When there is a mediator, people are interested in knowing the direct effect of the treatment on the outcome without passing the mediator and indirect effect via the mediator only (Xu et al., 2023), as illustrated in Fig. 3 and formally defined below, following the definitions in Pearl (2014).

**Definition 3.1** (Natural Direct Effect (NDE)). NDE $= \mathbb{E}[Y_i(t', M_i(t)) - Y_i(t, M_i(t))]$.

NDE quantifies the expected change in the outcome $Y_i$ when the treatment changes from $T_i = t$ to $T_i = t'$, keeping the mediator $M_i$ fixed at the value it would have taken under $T_i = t$. In the example in Fig. 3, it is the effect via the causal pathway $T_i \to Y_i$.

**Definition 3.2** (Natural Indirect Effect (NIE)). NIE $= \mathbb{E}[Y_i(t, M_i(t')) - Y_i(t, M_i(t))]$.

NIE measures the expected change in $Y_i$ when the treatment is held constant at $T_i = t$, while $M_i$ changes to the value it would have taken under $T_i = t'$. In the example in Fig. 3, it is the effect via the causal pathway $T_i \to M_i \to Y_i$.

Note that the choice of the baseline treatment level $t$ can vary depending on the specific context.

**Definition 3.3** (Total Effect (TE)). TE $= \mathbb{E}[Y_i(t', M_i(t')) - Y_i(t, M_i(t))]$.

TE measures the expected change in $Y_i$ when the treatment changes from $T_i = t$ to $T_i = t'$, while allowing the mediator $M_i$ to vary naturally under each treatment level.

We extend Definitions 3.1 and 3.2 to network data (Vander-Weele et al., 2012). Specifically, we instantiate these definitions in two distinct scenarios: vaccination and product promotion campaigns. As shown in Fig. 2(b), let $W_{t_i}$ and $W_{y_i}$ represent the aggregated treatment and the aggregated outcome of $i$'s neighbors respectively. We are interested in how the aggregated treatment directly and indirectly impacts $i$'s outcome, i.e. PDE and PIE respectively.

**Definition 3.4** (Peer Direct Effects (PDE)).

$$\text{PDE}(w'_t) = \mathbb{E}\left[Y_i(w'_{t_i}, W_{y_i}(w'_{t_i})) - Y_i(w_{t_i}, W_{y_i}(w'_{t_i}))\right], \tag{1a}$$

$$\text{PDE}(w_t) = \mathbb{E}\left[Y_i(w'_{t_i}, W_{y_i}(w_{t_i})) - Y_i(w_{t_i}, W_{y_i}(w_{t_i}))\right]. \tag{1b}$$

PDE represents the effect of changing the treatment status of an individual's neighbors while keeping $i$'s neighbors' outcome corresponding to a fixed treatment status. In the above definition of PDE, $W_{y_i}(w'_{t_i})$ represents the potential outcome when $W_{t_i}$ takes the value $w'_{t_i}$. $Y_i(w_{t_i}, W_{y_i}(w'_{t_i}))$ represents $i$'s potential outcome when $W_{t_i}$ is set to $w_{t_i}$ and $W_{y_i}$'s value corresponds to what it would be if $W_{t_i}$ were set to $w'_{t_i}$. The meanings of the remaining terms in the equation follow a similar logic.

**In the vaccination example**, corresponding to Eq. (1a): $w'_{t_i}$ means that neighbors of individual $i$ are vaccinated, and $w_{t_i}$ means that the neighbors are not vaccinated. PDE indicates the reduction in the risk of neighbors' infectiousness to individual $i$ due to their vaccination directly.

**In product promotion campaigns**, corresponding to Eq. (1b): $w'_{t_i}$ means neighbors of individual $i$ are exposed to an advertisement, and $w_{t_i}$ means that neighbors are not exposed to an advertisement. PDE indicates an increase in the likelihood of individual $i$'s purchasing a product due to its neighbors' exposure to an advertisement for the product.

**Definition 3.5** (Peer Indirect Effects (PIE)).

$$\text{PIE}(w'_t) = \mathbb{E}\left[Y_i(w_{t_i}, W_{y_i}(w'_{t_i})) - Y_i(w_{t_i}, W_{y_i}(w_{t_i}))\right], \tag{2a}$$

$$\text{PIE}(w_t) = \mathbb{E}\left[Y_i(w'_{t_i}, W_{y_i}(w'_{t_i})) - Y_i(w'_{t_i}, W_{y_i}(w_{t_i}))\right]. \tag{2b}$$

PIE represents the effect of changing the aggregated outcome of individual $i$'s neighbors on individual $i$'s outcome, while holding the neighbors' aggregated treatment constant.

**In the vaccination example**, corresponding to Eq. (2a), PIE captures the reduction of the risk of infection of individual $i$ from their neighbors due to the reduction of the risk of infection of the neighbors themselves resulting from their vaccination.

**In product promotion campaigns**, corresponding to Eq. (2b), PIE reflects the increase of the chance of purchase of the individual $i$ due to their neighbors' purchases resulting from their exposure in the promotional campaign.

**Proposition 3.6** (Peer Effect Decomposition (Baron & Kenny, 1986)). *The peer effect (PE) can be decomposed into the sum of the PDE and the PIE. That is,*

$$\begin{aligned} \text{PE} &= \mathbb{E}\left[Y_i(w'_{t_i}, W_{y_i}(w'_{t_i})) - Y_i(w_{t_i}, W_{y_i}(w_{t_i}))\right] \\ &= \text{PDE}(w'_t) + \text{PIE}(w'_t) \\ &= \text{PDE}(w_t) + \text{PIE}(w_t)\,. \end{aligned} \tag{3}$$

This proposition shows that the total peer effect integrates both the direct and indirect pathways of peer influence.

**Definition 3.7** (Self-Treatment Effect (STE)).

$$\text{STE} = \mathbb{E}\left[Y_i(t') - Y_i(t)\right]. \tag{4}$$

STE represents the effect when the treatment changes from $t$ to $t'$ of individual $i$.

**In the examples of vaccination**, STE indicates the average difference of potential outcomes of individuals if they had been vaccinated versus if they had not been vaccinated.

**In the examples of product promotion campaigns**, STE represents the average difference in individuals' potential outcomes between receiving the advertisement and not receiving it.

### 3.3. Problem Setting

The goal of this work is to estimate the causal effects PDE, PIE, and STE from network data. We provide a formal problem definition as follows.

**Problem Definition.** Given observational network data with node features, treatments, and outcomes, and assuming the causal structure as in Fig. 2b, our goal is to obtain unbiased estimation of the following causal effects.

- *Peer Direct Effects (PDE)*: The causal effect of $W_{t_i}$ on $Y_i$ via the pathway $W_{t_i} \to Y_i$, as formally defined in Definition 3.4.

- *Peer Indirect Effects (PIE)*: The causal effect of $W_{t_i}$ on $Y_i$ via the pathway $W_{t_i} \to W_{y_i} \to Y_i$, as formally defined in Definition 3.5.

- *Self-Treatment Effects (STE)*: The causal effect of $T_i$ on $Y_i$ via the pathway $T_i \to Y_i$, as formally defined in Definition 3.7.

As PE can be obtained based on PDE and PIE following proposition 3.6, we do not include the estimation of PE in our problem definition.

### 3.4. Assumptions

To estimate PDE, PIE, and STE from observational network data, some assumptions are required (Jiang & Sun, 2022).

**Assumption 3.8** (Network Unconfoundedness). The potential outcome is independent of both individual treatment and neighborhood exposure, given individual and neighbor features. i.e., $Y_i(t_i) \perp\!\!\!\perp T_i \mid \mathbf{X}_i, \{\mathbf{X}_j\}_{j \in \mathcal{N}_i}$ and $Y_i(w_{t_i}, w_{y_i}) \perp\!\!\!\perp W_{t_i} \mid \mathbf{X}_i, \{\mathbf{X}_j\}_{j \in \mathcal{N}_i}$.

**Assumption 3.9** (Network Consistency). The potential outcome equals the observed outcome when an individual is exposed to the same treatment and neighborhood exposure. i.e., $Y_i = Y_i(t_i)$ and $Y_i = Y_i(w_{t_i}, w_{y_i})$ when the individual is exposed at $t_i$ and $(w_{t_i}, w_{y_i})$, respectively.

**Assumption 3.10** (Network Overlap). Every treatment and neighborhood exposure combination $(T_i, W_{t_i}, W_{y_i})$ must have a positive probability of occurring; i.e., $0 <$

$\mathbb{P}(t_i \mid \mathbf{X}_i, \{\mathbf{X}_j\}_{j \in \mathcal{N}_i}) < 1$ and $0 < \mathbb{P}(w_{t_i}, w_{y_i} \mid \mathbf{X}_i, \{\mathbf{X}_j\}_{j \in \mathcal{N}_i}) < 1$.

These assumptions collectively ensure that causal effects can be estimated from observational network data.

## 4. The Proposed gDIS Method

In this section, we present the theoretical analysis of the identifiability of PDE, PIE, and STE, and introduce our gDIS model for estimating these effects from network data.

### 4.1. Identifiability of PDE, PIE, and STE

To present our main theoretical results (Lemma 4.1 and Theorem 4.2), we need the following graphical criterion for identifying an appropriate adjustment set relevant to the identifiability of the PDE and PIE.

**Criterion 1** (Sequential Ignorability (Pearl, 2014; Imai et al., 2010)). *A set of observed covariates* $\mathbf{W}$ *is said to satisfy sequential ignorability if it blocks all back-door paths from* $M_i$ *to* $Y_i$ *that do not pass through* $T_i$ ($T_i$-*avoiding backdoor paths), and all back-door paths from* $T_i$ *to either* $M_i$ *or* $Y_i$, *and no element of* $\mathbf{W}$ *is a descendant of* $T_i$.

We present our first theoretical finding below.

**Lemma 4.1.** *In the causal DAG represented in Fig. 2(b), the aggregated set of neighbor features* $\mathbf{W}_{x_i}$ *satisfies Criterion 1.*

*Proof.* First, $\mathbf{W}_{x_i}$ blocks all backdoor paths from $W_{y_i}$ to $Y_i$ that do not pass through $W_{t_i}$. In the causal DAG shown in Fig. 2(b), all $W_{t_i}$-avoiding backdoor paths from $W_{y_i}$ to $Y_i$ ($W_{y_i} \leftarrow \mathbf{W}_{x_i} \to Y_i$, $W_{y_i} \leftarrow \mathbf{W}_{x_i} \to T_i \leftarrow \mathbf{X}_i \to Y_i$, and $W_{y_i} \leftarrow \mathbf{W}_{x_i} \to T_i \to Y_i$) are blocked by the set $\mathbf{W}_{x_i}$. Next, from Fig. 2(b), there are the following backdoor paths from $W_{t_i}$ to $W_{y_i}$ ($W_{t_i} \leftarrow \mathbf{W}_{x_i} \to W_{y_i}$, $W_{t_i} \leftarrow \mathbf{W}_{x_i} \to Y_i \leftarrow W_{y_i}$, $W_{t_i} \leftarrow \mathbf{W}_{x_i} \to T_i \to Y_i \leftarrow W_{y_i}$, and $W_{t_i} \leftarrow \mathbf{W}_{x_i} \to T_i \leftarrow \mathbf{X}_i \to Y_i \leftarrow W_{y_i}$) and from $W_{t_i}$ to $Y_i$ ($W_{t_i} \leftarrow \mathbf{W}_{x_i} \to Y_i$, $W_{t_i} \leftarrow \mathbf{W}_{x_i} \to W_{y_i} \to Y_i$, $W_{t_i} \leftarrow \mathbf{W}_{x_i} \to T_i \to Y_i$, and $W_{t_i} \leftarrow \mathbf{W}_{x_i} \to T_i \leftarrow \mathbf{X}_i \to Y_i$). We can see that $\mathbf{W}_{x_i}$ blocks all these backdoor paths. Finally, $\mathbf{W}_{x_i}$ is not a descendant of $W_{t_i}$ in Fig. 2(b). Therefore, $\mathbf{W}_{x_i}$ satisfies the Criterion 1 and thus can serve as an adjustment set for unbiased estimation of PDE and PIE. $\square$

Based on Lemma 4.1 and Eq.13 in (Pearl, 2014), the PDE (1a), PIE (2a), and STE (4) can be represented using do-expressions. Furthermore, using Pearl's back-door adjustment formula and do-calculus rules (Pearl, 2009), these do-expressions can be further transformed into probability expressions. This result forms the basis of our theoretical finding in Theorem 4.2.

**Theorem 4.2.** *If* $\mathbb{P}(T_i, Y_i, \mathbf{X}_i, \mathbf{W}_{x_i}, W_{y_i}, W_{t_i})$ *is generated from the causal DAG in Fig. 2(b), then the PDE, PIE, and STE can be identified from the data as follows:*

$$
\begin{aligned}
\text{PDE} = &\big[\mathbb{E}(Y_i \mid W_{t_i} = w'_{t_i}, W_{y_i} = w'_{y_i}, \mathbf{W}_{x_i} = \mathbf{w}_{x_i}) \\
&- \mathbb{E}(Y_i \mid W_{t_i} = w_{t_i}, W_{y_i} = w'_{y_i}, \mathbf{W}_{x_i} = \mathbf{w}_{x_i})\big] \\
&\times \mathbb{P}(W_{y_i} = w'_{y_i} \mid W_{t_i} = w'_{t_i}, \mathbf{W}_{x_i} = \mathbf{w}_{x_i}) \\
&\times \mathbb{P}(\mathbf{W}_{x_i} = \mathbf{w}_{x_i}),
\end{aligned} \tag{5}
$$

$$
\begin{aligned}
\text{PIE} = &\mathbb{E}(Y_i \mid W_{t_i} = w_{t_i}, W_{y_i} = w_{y_i}, \mathbf{W}_{x_i} = \mathbf{w}_{x_i}) \\
&\times \big[\mathbb{P}(W_{y_i} = w_{y_i} \mid W_{t_i} = w'_{t_i}, \mathbf{W}_{x_i} = \mathbf{w}_{x_i}) \\
&- \mathbb{P}(W_{y_i} = w_{y_i} \mid W_{t_i} = w_{t_i}, \mathbf{W}_{x_i} = \mathbf{w}_{x_i})\big] \\
&\times \mathbb{P}(\mathbf{W}_{x_i} = \mathbf{w}_{x_i}),
\end{aligned} \tag{6}
$$

$$
\begin{aligned}
\text{STE} = &\mathbb{P}(\mathbf{X}_i = \mathbf{x}_i)\mathbb{P}(\mathbf{W}_{x_i} = \mathbf{w}_{x_i}) \\
&\times \big[\mathbb{E}(Y_i \mid T_i = t'_i, \mathbf{X}_i = \mathbf{x}_i, \mathbf{W}_{x_i} = \mathbf{w}_{x_i}) \\
&- \mathbb{E}(Y_i \mid T_i = t_i, \mathbf{X}_i = \mathbf{x}_i, \mathbf{W}_{x_i} = \mathbf{w}_{x_i})\big].
\end{aligned} \tag{7}
$$

*Proof.* When Lemma 4.1 is satisfied, PDE, PIE and STE are represented as the following in do-expression (Pearl, 2014). The analogous representations for PDE (1b) and PIE (2b) are omitted for brevity.

$$
\begin{aligned}
\text{PDE} = &\mathbb{E}\big[Y_i \mid do(W_{t_i} = w'_{t_i}, W_{y_i} = w'_{y_i}), \mathbf{W}_{x_i} = \mathbf{w}_{x_i}\big] \\
&- \mathbb{E}\big[Y_i \mid do(W_{t_i} = w_{t_i}, W_{y_i} = w'_{y_i}), \mathbf{W}_{x_i} = \mathbf{w}_{x_i}\big] \\
&\times \mathbb{P}\big(W_{y_i} = w'_{y_i} \mid do(W_{t_i} = w'_{t_i}), \mathbf{W}_{x_i} = \mathbf{w}_{x_i}\big) \\
&\times \mathbb{P}\big(\mathbf{W}_{x_i} = \mathbf{w}_{x_i}\big),
\end{aligned} \tag{8}
$$

$$
\begin{aligned}
\text{PIE} = &\mathbb{E}\big(Y_i \mid do(W_{t_i} = w_{t_i}, W_{y_i} = w_{y_i}), \mathbf{W}_{x_i} = \mathbf{w}_{x_i}\big) \\
&\times \big[\mathbb{P}(W_{y_i} = w_{y_i} \mid do(W_{t_i} = w'_{t_i}), \mathbf{W}_{x_i} = \mathbf{w}_{x_i}) \\
&- \mathbb{P}(W_{y_i} = w_{y_i} \mid do(W_{t_i} = w_{t_i}), \mathbf{W}_{x_i} = \mathbf{w}_{x_i})\big] \\
&\times \mathbb{P}(\mathbf{W}_{x_i} = \mathbf{w}_{x_i}),
\end{aligned} \tag{9}
$$

$$
\text{STE} = \mathbb{E}\left(Y_i \mid do(T_i = t'_i)\right) - \mathbb{E}\left(Y_i \mid do(T_i = t_i)\right). \tag{10}
$$

We now prove that Eq. (8) to Eq. (10) are identifiable, i.e., the do-operators can be reduced to the do-free expression in Eq. (5) to Eq. (7), respectively. The causal DAG is shown in Fig 2(b). Based on rule 2 of do calculus (Pearl, 2009), we have $\mathbb{P}(Y_i = y_i \mid do(W_{t_i} = w'_{t_i}, W_{y_i} = w'_{y_i}), \mathbf{W}_{x_i} = \mathbf{w}_{x_i}) = \mathbb{P}(Y_i = y_i \mid do(W_{t_i} = w'_{t_i}), W_{y_i} = w'_{y_i}, \mathbf{W}_{x_i} = \mathbf{w}_{x_i})$ since $(Y_i \perp\!\!\!\perp W_{y_i} \mid W_{t_i}, \mathbf{W}_{x_i})_{G_{\overline{W_{t_i}}\underline{W_{y_i}}}}$, where $\overline{W_{t_i}}$ represents the removing all edges with arrows pointing to $W_{t_i}$, and $\underline{W_{y_i}}$ represents the removing all edges with arrows emanating from $W_{y_i}$.

Continue the above reduction, we have $\mathbb{P}(Y_i = y_i \mid do(W_{t_i} = w'_{t_i}), W_{y_i} = w'_{y_i}, \mathbf{W}_{x_i} = \mathbf{w}_{x_i}) = \mathbb{P}(Y_i = y_i \mid W_{t_i} = w'_{t_i}, W_{y_i} = w'_{y_i}, \mathbf{W}_{x_i} = \mathbf{w}_{x_i})$ because

$(Y_i \perp\!\!\!\perp W_{t_i} \mid W_{y_i}, \mathbf{W}_{x_i})_{G_{\underline{W_{t_i}}}}$, where $\underline{W_{t_i}}$ represents the removing all edges with arrows emanating from $W_{t_i}$ (rule 2 of do calculus).

We apply rule 2 of do calculus again. $\mathbb{P}(W_{y_i} = w'_{y_i} \mid do(W_{t_i} = w'_{t_i}), \mathbf{W}_{x_i} = \mathbf{w}_{x_i}) = \mathbb{P}(W_{y_i} = w'_{y_i} \mid W_{t_i} = w'_{t_i}, \mathbf{W}_{x_i} = \mathbf{w}_{x_i})$ because $(W_{y_i} \perp\!\!\!\perp W_{t_i} \mid \mathbf{W}_{x_i})_{G_{\underline{W_{t_i}}}}$.

Up to this point, Eq. (8) has been reduced to Eq. (5). The proof for reducing Eq. (9) to Eq. (6) is similar, and we do not repeat the derivations here.

We now reduce Eq. (10) to Eq. (7). Based on the back-door adjustment formula (Pearl, 2009), $\mathbb{P}(Y_i = y_i \mid do(T_i = t'_i))$ is identifiable because all backdoor paths from $T_i$ to $Y_i$ are blocked by $\mathbf{X}_i$ and $\mathbf{W}_{x_i}$. Specifically, $T_i \leftarrow \mathbf{X}_i \rightarrow Y_i$ is blocked by $\mathbf{X}_i$, $T_i \leftarrow \mathbf{W}_{x_i} \rightarrow W_{t_i} \rightarrow Y_i$ is blocked by $\mathbf{W}_{x_i}$, $T_i \leftarrow \mathbf{W}_{x_i} \rightarrow W_{t_i} \rightarrow W_{y_i} \rightarrow Y_i$ is blocked by $\mathbf{W}_{x_i}$, $T_i \leftarrow \mathbf{W}_{x_i} \rightarrow W_{y_i} \rightarrow Y_i$ is blocked by $\mathbf{W}_{x_i}$, and $T_i \leftarrow \mathbf{W}_{x_i} \rightarrow W_{y_i} \leftarrow W_{t_i} \rightarrow Y_i$ is blocked by $\mathbf{W}_{x_i}$. Hence, $\mathbb{P}(Y_i = y_i \mid do(T_i = t'_i)) = \mathbb{P}(Y_i = y_i \mid T_i = t'_i, \mathbf{X}_i = \mathbf{x}_i, \mathbf{W}_{x_i} = \mathbf{w}_{x_i})\mathbb{P}(\mathbf{X}_i = \mathbf{x}_i)\mathbb{P}(\mathbf{W}_{x_i} = \mathbf{w}_{x_i})$. $\square$

### 4.2. The Proposed gDIS Model

In this section, we present the gDIS model for estimating PDE, PIE and STE from observational network data by following Eq. (5) to Eq. (7). Due to space limit the workflow of the model is shown in Fig. 6 in Appendix C. Calculating the posterior probability of $W_{y_i}$ and $Y_i$ is challenging using traditional methods due to the non-linear relationships between $\mathbf{W}_{x_i}$ and both $W_{y_i}$ and $Y_i$, the continuous nature of $W_{y_i}$, and the high-dimensional space of $\mathbf{W}_{x_i}$. GNNs have demonstrated an outstanding capability to capture such non-linear relationships in network data (Scarselli et al., 2008; Kipf & Welling, 2017). Thus, we propose a practical solution combining a GNN with attention mechanisms and a fully connected layer. The GNN models node interactions and extracts higher-order structural information, while the fully connected layer captures non-linear relationships between $\mathbf{W}_{x_i}$ and $W_{y_i}$. This design enables the gDIS model to estimate causal effects accurately in complex networks.

To estimate PDE, PIE and STE as identified in Eq. (5) to Eq. (7), respectively, we need to compute the aggregated variables: $W_{t_i} = \sum_{j \in \mathcal{N}_i} w_{ij} T_j$, $W_{y_i} = \sum_{j \in \mathcal{N}_i} w_{ij} Y_j$, and $\mathbf{W}_{x_i} = \sum_{j \in \mathcal{N}_i} w_{ij} \mathbf{X}_j$, where $w_{ij} = \frac{\exp(e_{ij})}{\sum_{k \in \mathcal{N}_i} \exp(e_{ik})}$, $e_{ij} = \frac{\mathbf{X}_i \cdot \mathbf{X}_j}{\|\mathbf{X}_i\|\|\mathbf{X}_j\|}$, and $\mathcal{N}_i$ represents the set of neighbors of node $i$ in the network. Correspondingly, as shown in Fig. 6 in Appendix C and Algorithm 1 in Appendix D, gDIS consists of three key stages as detailed in the following.

**Stage 1**: To estimate the conditional probability $P(W_{y_i} \mid W_{t_i}, \mathbf{W}_{x_i})$, we employ a 2-layer GNN with attention mechanisms and a single fully connected layer as the es-

timator. The estimator, denoted as $f_1 : (W_{t_i}, \mathbf{W}_{x_i}) \to W_{y_i}$, is designed to model this mapping by minimizing the mean squared error (MSE) (Chicco et al., 2021). The loss function is defined as:

$$\mathcal{L}_1 = \frac{1}{m} \sum_{i=1}^{m} (W_{y_i} - f_1(W_{t_i}, \mathbf{W}_{x_i}))^2. \qquad (11)$$

**Stage 2**: To estimate the conditional probability $P(Y_i | W_{t_i}, W_{y_i}, \mathbf{W}_{x_i})$, we use a 2-layer GNN with attention mechanisms and a 3-layer fully connected network as the estimator to refine predictions. The estimator, denoted as $f_2 : (W_{t_i}, W_{y_i}, \mathbf{W}_{x_i}) \to Y_i$, models this mapping by optimizing the total loss function, defined as:

$$\begin{aligned} \mathcal{L}_2 = & \frac{1}{m} \sum_{i=1}^{m} (Y_i - f_2(W_{t_i}, W_{y_i}, \mathbf{W}_{x_i}))^2 \\ & + \lambda \cdot \frac{1}{(m-1)^2} \text{trace}(\mathbf{K_H C K_{H'} C}). \end{aligned} \qquad (12)$$

The first term represents the MSE for prediction accuracy, and the second term is the Hilbert-Schmidt Independence Criterion (HSIC) regularization. The hyperparameter $\lambda$ balances the HSIC term's contribution. $\mathbf{C} = \mathbf{I} - \frac{1}{m}\mathbf{1}\mathbf{1}^T$ is the centering matrix that removes the mean from the kernel matrices, $\mathbf{I}$ is the $m \times m$ identity matrix, $\mathbf{1}$ is an $m \times 1$ vector of ones.

The HSIC regularization measures the dependence between node features $\mathbf{X}$ and the learned embeddings in a Reproducing Kernel Hilbert Space (RKHS), ensuring the embeddings do not retain spurious correlations from the original features, thereby enhancing model robustness (Berlinet & Thomas-Agnan, 2011). HSIC is computed by first defining similarity matrices for the feature space ($\mathbf{H}$) and the embedding space ($\mathbf{H'}$) using Gaussian kernels:

$$(\mathbf{K_H})_{il} = \exp(-\frac{\|\mathbf{H}_i - \mathbf{H}_l\|^2}{2\gamma^2}), \qquad (13)$$

$$(\mathbf{K_{H'}})_{il} = \exp(-\frac{\|\mathbf{H}'_i - \mathbf{H}'_l\|^2}{2\gamma^2}). \qquad (14)$$

where $\gamma$ is the Gaussian kernel bandwidth (Kakde et al., 2017), set to the median of input feature distances.

**Stage 3**: Kernel density estimation (KDE) is applied to density estimation in high-dimensional settings (Botev et al., 2010; Duong & Hazelton, 2003). We employ KDE to estimate the probability density functions of $P(\mathbf{X}_i)$ and $P(\mathbf{W}_{x_i})$.

$$\log P(\mathbf{X}_i) = \log(\frac{1}{m}\sum_{k=1}^{n}\frac{1}{\sqrt{2\pi}h}\exp(-\frac{\|\mathbf{X}_i - \mathbf{X}_k\|^2}{2h^2})), \qquad (15)$$

$$\log P(\mathbf{W}_{x_i}) =$$
$$\log(\frac{1}{m}\sum_{k=1}^{n}\frac{1}{\sqrt{2\pi}h}\exp(-\frac{\|\mathbf{W}_{x_i} - \mathbf{W}_{x_k}\|^2}{2h^2})). \qquad (16)$$

where $h$ is the bandwidth parameter in KDE, which controls the smoothness of the estimated density.

After obtaining the estimation of the conditional probabilities and the marginal probabilities, PDE, PIE and STE can be estimated via Eq. (5) to Eq. (7) respectively.

## 5. Experiments

In this section, we first evaluate the performance of gDIS on real-world social network datasets and compare it with existing methods. Then, we assess gDIS under different hyperparameter configurations and analyze its time complexity. Finally, we conduct a case study using a recommendation dataset to evaluate the effectiveness of disentangling peer effects into direct and indirect components. For each experiment, we repeat it five times and report the average result and the standard deviation. Detailed experimental setup, parameters, and model architectures are provided in Appendix G.

**Baselines.** We compared our model with six baselines: (1) CFR+(N) (Shalit et al., 2017), which estimates effects and incorporates peer exposure using integral probability metrics (IPM) for distribution balancing. (2) TARNet+(N) (Shalit et al., 2017), a variant of CFR+(N) that does not use IPM. (3) NetDeconf (Guo et al., 2020), an adaptation of CFR for network data that uses GNNs to encode confounders. (4) 1-GNN (Ma & Tresp, 2021), a GNN-based causal effect estimator for network interference. (5) NetEst (Jiang & Sun, 2022), which applies adversarial learning to bridge graph machine learning and causal effect estimation. (6) TNet (Chen et al., 2024) integrates target learning to improve the accuracy of effect estimation. (7) gDIS(-HSIC), our gDIS method without the HSIC module.

**Metrics.** We evaluate the algorithms using MSE (Chicco et al., 2021) and PEHE (Grimmer et al., 2017). MSE ($\epsilon_{MSE} = \frac{1}{m}\sum_{i=1}^{m}(\hat{Y}_i - Y_i)^2$) measures the bias in effect estimation, while PEHE ($\epsilon_{PEHE} = \sqrt{\frac{1}{m}\sum_{i=1}^{m}\left[(\hat{Y}_i(t') - \hat{Y}_i(t)) - (Y_i(t') - Y_i(t))\right]^2}$) quantifies the precision of the effect estimation. $m$ is the number of samples, and $\hat{Y}_i$ and $Y_i$ denote the predicted and true outcomes for individual $i$, respectively. Lower values correspond to better performance.

*Table 2.* Causal effect estimation results. The $\epsilon_{PEHE}$ error is reported. The top-performing results are emphasized in bold, and the second-best results are underlined. Note that " / " indicates the model is not applicable for this effect.

| Data | Effects | CFR(+N) | ND(+N) | TARNET(+N) | 1-GNN | NetEst | TNet | gDIS(-HSIC) | gDIS |
|---|---|---|---|---|---|---|---|---|---|
| BC (within-sample) | peer | $0.7412_{\pm0.0112}$ | $0.7545_{\pm0.0121}$ | $0.6991_{\pm0.014}$ | $0.6770_{\pm0.0340}$ | $0.6729_{\pm0.0326}$ | $0.6591_{\pm0.0362}$ | $0.2478_{\pm0.0197}$ | $\mathbf{0.2093_{\pm0.014}}$ |
| | peer direct | / | / | / | / | / | / | $\underline{0.1456_{\pm0.0129}}$ | $\mathbf{0.1265_{\pm0.0126}}$ |
| | peer indirect | / | / | / | / | / | / | $\underline{0.194_{\pm0.0018}}$ | $\mathbf{0.1358_{\pm0.0026}}$ |
| | self-treatment | $0.6059_{\pm0.0263}$ | $0.4648_{\pm0.0353}$ | $0.4282_{\pm0.0326}$ | $0.3600_{\pm0.0330}$ | $0.4353_{\pm0.0323}$ | $0.462_{\pm0.0273}$ | $\underline{0.0673_{\pm0.0025}}$ | $\mathbf{0.0588_{\pm0.0015}}$ |
| BC (out-of-sample) | peer | $0.7399_{\pm0.0108}$ | $0.7549_{\pm0.0122}$ | $0.6094_{\pm0.0122}$ | $0.6860_{\pm0.0275}$ | $0.6823_{\pm0.0260}$ | $0.657_{\pm0.0248}$ | $0.2476_{\pm0.0128}$ | $\mathbf{0.2034_{\pm0.0124}}$ |
| | peer direct | / | / | / | / | / | / | $0.1576_{\pm0.0101}$ | $\mathbf{0.1101_{\pm0.014}}$ |
| | peer indirect | / | / | / | / | / | / | $0.1979_{\pm0.002}$ | $\mathbf{0.1312_{\pm0.0029}}$ |
| | self-treatment | $0.6062_{\pm0.0262}$ | $0.4588_{\pm0.0340}$ | $0.4240_{\pm0.0226}$ | $0.3672_{\pm0.0315}$ | $0.4425_{\pm0.0304}$ | $0.4584_{\pm0.0253}$ | $\underline{0.0656_{\pm0.0034}}$ | $\mathbf{0.0545_{\pm0.0023}}$ |
| Flickr (within-sample) | peer | $0.9736_{\pm0.0078}$ | $0.9820_{\pm0.0116}$ | $0.9751_{\pm0.0079}$ | $0.9890_{\pm0.0130}$ | $0.9650_{\pm0.0078}$ | $0.9757_{\pm0.0430}$ | $0.1312_{\pm0.0007}$ | $\mathbf{0.1262_{\pm0.0007}}$ |
| | peer direct | / | / | / | / | / | / | $0.1138_{\pm0.0013}$ | $\mathbf{0.1118_{\pm0.0013}}$ |
| | peer indirect | / | / | / | / | / | / | $0.0933_{\pm0.0008}$ | $\mathbf{0.0901_{\pm0.0008}}$ |
| | self-treatment | $0.5249_{\pm0.0115}$ | $0.5293_{\pm0.0193}$ | $0.4740_{\pm0.0129}$ | $0.3870_{\pm0.0130}$ | $0.3588_{\pm0.0125}$ | $0.3696_{\pm0.0901}$ | $\underline{0.1599_{\pm0.003}}$ | $\mathbf{0.1578_{\pm0.003}}$ |
| Flickr (out-of-sample) | peer | $0.9845_{\pm0.0068}$ | $0.9830_{\pm0.0112}$ | $0.9848_{\pm0.0080}$ | $0.9840_{\pm0.0113}$ | $0.9651_{\pm0.0078}$ | $0.9762_{\pm0.0730}$ | $0.1307_{\pm0.0007}$ | $\mathbf{0.1206_{\pm0.0007}}$ |
| | peer direct | / | / | / | / | / | / | $0.1109_{\pm0.0011}$ | $\mathbf{0.1008_{\pm0.0012}}$ |
| | peer indirect | / | / | / | / | / | / | $0.0898_{\pm0.0008}$ | $\mathbf{0.086_{\pm0.0008}}$ |
| | self-treatment | $0.5242_{\pm0.0109}$ | $0.52843_{\pm0.0186}$ | $0.5580_{\pm0.0112}$ | $0.3850_{\pm0.0085}$ | $0.3409_{\pm0.0105}$ | $0.3703_{\pm0.0843}$ | $\underline{0.1543_{\pm0.0032}}$ | $\mathbf{0.1502_{\pm0.0033}}$ |

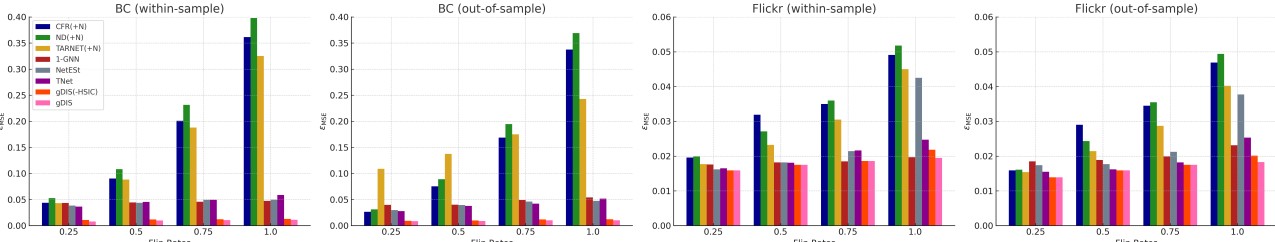

*Figure 4.* The results illustrating the relationship between the estimation bias ($\epsilon_{MSE}$) and the percentage of units with treatment flip.

## 5.1. Performance Evaluation on gDIS with Social Network Data

We applied gDIS to two semi-synthetic real-world datasets, BlogCatalog (BC) and Flickr (Li et al., 2015), to evaluate its effectiveness on estimating PDE, PIE and STE. Due to space limitations, details about the data preprocessing and generation process are provided in Appendix E. We evaluate the method using "within-sample" estimates on training networks and "out-of-sample" estimates on testing networks.

**Causal Effect Estimation Results.** Table 2 shows the $\epsilon_{PEHE}$ results on the BC and Flickr datasets. The complete gDIS model demonstrates superior performance in both within-sample and out-of-sample scenarios, outperforming existing baselines. This advantage arises from our explicit modeling of neighbor influences under natural network conditions. By effectively leveraging graph structure and disentangling peer effects into direct and indirect components, gDIS provides a more comprehensive framework for causal inference in social networks.

**The role of HSIC.** To validate the effectiveness of HSIC regularization, we implemented a variant of gDIS without the HSIC component (denoted as gDIS(-HSIC) in Table 2). From Table 2, we observe that the full version of gDIS outperforms the variant without HSIC. This result implies that HSIC can promote independence between node features and learned embeddings, thereby enhancing model robustness and mitigating overfitting.

**Bias Analysis with Simulated Treatment Flip.** To evaluate the robustness of gDIS under treatment assignment noise, we follow (Jiang & Sun, 2022) and simulate outcomes with varying rates of treatment flips (0.25, 0.5, 0.75, 1). As shown in Fig. 4, higher flip rates introduce more noise, increasing MSE. gDIS shows lower MSE on Flickr than that on BC, likely due to the richer connectively in the Flickr dataset. Across all flip rates, gDIS consistently outperforms other methods, demonstrating strong resilience to noisy treatment assignments.

**Hyperparameter Impact.** We analyzed the impact of the HSIC regularization coefficient $\lambda$, varying from 0 to 0.5, on the error $\epsilon_{PEHE}$, as shown in Fig. 5. We found that $\lambda = 0.3$ produced the best overall performance. When $\lambda > 0.3$, it begins to over-penalize the feature representations, leading to a decline in performance.

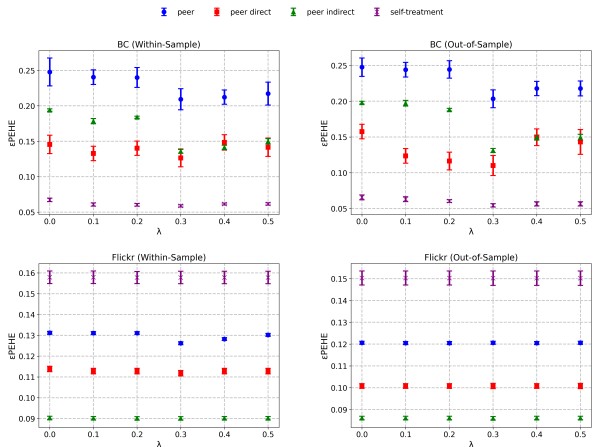

*Figure 5.* Impact of HSIC regularization ($\lambda$) on $\epsilon_{PEHE}$ for BC and Flickr datasets across different effect types.

**Time Complexity.** The time complexity analysis is provided in Appendix H.

### 5.2. A Case Study of gDIS for Recommendation Systems

Understanding the impact of PDE and PIE is crucial for designing effective interventions in recommendation systems. To highlight the importance of separating PDE from PIE, we conducted experiments using the Coat Dataset[1], which simulates MNAR (missing not at random) data from online coat purchases. As described in (Schnabel et al., 2016), the dataset includes user features and 5-point ratings, with 1 being the lowest preference. Each user-item interaction with one of the top 10 rated products is marked as ad exposure, i.e., treatment = 1, and a rating of 4 or higher implies a purchase. The user-level treatment is calculated as the proportion of interactions with promoted items.

User relationships are constructed using pairwise cosine similarity between feature vectors, which represents the strength of the relationships. This network structure enables us to measure how ad exposure spreads through social connections and influences purchasing behavior. The estimated causal effects in this network are as follows: PDE = 0.0127, PIE = 0.0212, PE = 0.0339, and STE = 0.0635.

To better understand the relative importance of different causal mechanisms, we analyzed the proportional contributions of STE, PDE, and PIE to the total effect, i.e., PE + STE = 0.0974. STE contributes 65.20% of the total effect, while network effects account for 34.80%, including 21.76% from PIE, notably larger than the contribution of PDE (13.04%). This distribution provides key insights for recommendation system design. The dominance of STE suggests that di-

rect user targeting should receive the majority of marketing resources, while the substantial network effect proportion indicates social influence strategies remain important. The fact that PIE exceeds PDE highlights that users are more influenced by their peers' actual purchasing behaviors than by peer ad exposure alone. Recommendation strategies should therefore focus not only on expanding ad reach but also on optimizing early adopters' satisfaction to maximize cascading effects through social networks. Leveraging these patterns can significantly enhance campaign effectiveness by strategically amplifying indirect peer effects.

## 6. Conclusion

**Summary of Contributions.** In this work, we address the novel problem of differentiating and estimating the three types of causal effects PDE, PIE and STE in observational network data. Through theoretical analysis, we establish the identifiability conditions for estimating these causal effects from the data and provide corresponding proofs for the identifiability of these causal effects. Supported by the identification results, we propose the gDIS algorithm for accurately estimating PDE, PIE, and STE in observational network data. To capture complex network interactions, gDIS employs a multi-layer GNN with attention mechanisms and incorporates HSIC to effectively reduce dependencies between node features. We have validated the effectiveness and robustness of gDIS on two semi-synthetic social network datasets and a recommendation system dataset. gDIS has demonstrated strong performance even in complex network environments.

**Limitations & Future Work.** Although supported by the theoretical and empirical results, gDIS has a major limitation. It relies on the network unconfoundedness assumption, which may be violated in practice, despite being commonly seen in the literature. Future work will focus on relaxing this assumption to enhance its applicability by exploring approaches like instrumental variables (IVs) to mitigate hidden confounders, network representation learning to capture latent structures, and causal discovery methods to identify potential confounders.

## Acknowledgements

This work has been partially supported by the Australian Research Council (grant number DP230101122).

## Impact Statement

This work addresses the critical challenge of differentiating and estimating the three types of causal effects PDE, PIE, and STE in observational network data. Through theoretical analysis, we establish identifiability conditions for these

---

[1]https://www.cs.cornell.edu/~schnabts/mnar/

causal effects and provide corresponding proofs, laying a solid foundation for causal inference in networks. Building on these results, we propose the gDIS framework, which integrates graph neural networks, attention mechanisms, and HSIC regularization to effectively estimate PDE, PIE, and STE. This contribution is potentially useful for optimizing intervention strategies in various applications and domains such as public health, marketing, and social influence analysis.

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

# Supplement to "Telling Peer Direct Effects from Indirect Effects in Observational Network Data"

## A. Background on Causal Inference

To aid readers' understanding, in the following, we define the terms causal DAG, path, d-separation, Structural Causal Model, and Back-door Criterion, which are used throughout this paper.

**Causal DAG**. Let $\mathcal{G} = (\mathbf{V}, \mathbf{E})$ be a directed acyclic graph (DAG), where $\mathbf{V} = \{V_1, \dots, V_m\}$ and $\mathbf{E} \subseteq \mathbf{V} \times \mathbf{V}$ are the set of nodes and edges of the DAG respectively. $\mathcal{G}$ is a causal DAG if each directed edge $V_i \to V_j \in \mathbf{E}$ encodes a direct causal relationship of $V_i$ on $V_j$.

**Path**. A *path* $\pi$ between nodes $V_i$ and $V_j$ in $\mathcal{G}$ is any sequence of distinct nodes or variables $\langle V_i, \dots, V_j \rangle$ such that each consecutive pair is adjacent (i.e., there is an edge between the pair of nodes). If every edge on $\pi$ points toward $V_j$, then $\pi$ is called a *directed* (or *causal*) path; in this case $V_i$ is an *ancestor* of $V_j$ and $V_j$ a *descendant* of $V_i$.

**d-Separation**. In the DAG $\mathcal{G} = (\mathbf{V}, \mathbf{E})$, a path $\pi$ between $V_i$ and $V_j$ is said to be *blocked* (or *d-separated*) by a set $\mathbf{M} \subseteq \mathbf{V}$ if (i) $\pi$ contains a chain $V_a \to V_b \to V_c$ or a fork $V_a \leftarrow V_b \to V_c$ with $V_b \in \mathbf{M}$, and (ii) $\pi$ contains a collider $V_a \to V_b \leftarrow V_c$ such that neither $V_b$ nor any of its descendants is in $\mathbf{M}$. Otherwise $V_i$ and $V_j$ are *d-connected* given $\mathbf{M}$. We write $V_i \perp\!\!\!\perp V_j \mid \mathbf{M}$ when in $\mathcal{G}$ every path between $V_i$ and $V_j$ is blocked by $\mathbf{M}$.

**Structural Causal Model**. A Structural Causal Model is a tuple $\mathcal{M} = (\mathbf{U}, \mathbf{V}, \mathbf{F})$, where $\mathbf{U} = \{U_1, \dots, U_m\}$ are exogenous variables, $\mathbf{V} = \{V_1, \dots, V_m\}$ are endogenous variables, and $\mathbf{F} = \{f_i\}$ is a collection of structural functions $f_i \colon \mathrm{Pa}(V_i) \times U_i \to V_i$, where $\mathrm{Pa}(V_i)$ is the set of direct causes of $V_i$.

**The Back-door Criterion** (Pearl, 2009). In a DAG $\mathcal{G} = (\mathbf{V}, \mathbf{E})$, for an ordered pair $(V_i, V_j)$, a set $\mathbf{M} \subseteq \mathbf{V} \setminus \{V_i, V_j\}$ is called a *back-door set* relative to $(V_i, V_j)$ if no variable in $\mathbf{M}$ is a descendant of $V_i$ and $\mathbf{M}$ blocks every path from $V_i$ to $V_j$ that begins with an arrow into $V_i$.

## B. Symbols

A summary of the key symbols used in the paper is presented in Table 3.

*Table 3.* Key symbols used in the paper

| Symbol | Description |
|---|---|
| $\mathbf{X}_i$ | Features vector of individual $i$ |
| $\mathrm{T}_i$ | Treatment of individual $i$ |
| $\mathrm{Y}_i$ | Outcome of individual $i$ |
| $\mathbf{W}_{x_i}$ | Aggregated neighbor features for individual $i$ |
| $W_{t_i}$ | Aggregated neighbor treatment exposure for individual $i$ |
| $W_{y_i}$ | Aggregated neighbor outcome exposure for individual $i$ |
| $W_{y_i}(w'_{t_i})$ | The potential outcome of $W_{y_i}$ when the treatment $W_{t_i}$ takes value $w'_{t_i}$ |
| $Y_i(w_{t_i}, W_{y_i}(w'_{t_i}))$ | The potential outcome of $Y_i$ when $W_{t_i}$ takes value $w_{t_i}$ and $W_{y_i}$ is fixed at what it would be when its treatment takes value $w'_{t_i}$ |
| k | Dimension of the feature vector of a node |
| $\mathcal{N}_i$ | Neighbors set of individual $i$ |
| m | Sample number |

## C. Workflow of the gDIS Framework

Fig. 6 illustrates the complete workflow of the gDIS model for estimating PDE, PIE, and STE in network data. The process consists of three main stages: the first stage focuses on estimating $W_{y_i}$, the second stage incorporates HSIC regularization

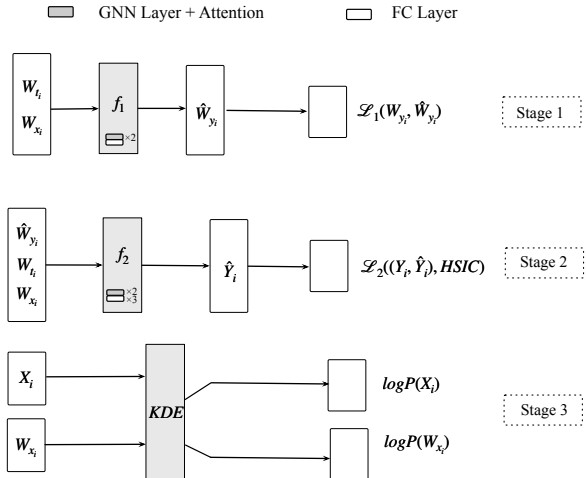

*Figure 6.* The workflow of our gDIS model for estimating PDE, PIE, PE and STE within network data.

for response prediction, and the third stage computes the probability density functions $\mathbb{P}(\mathbf{X}_i)$ and $\mathbb{P}(\mathbf{W}_{x_i})$ using kernel density estimation.

## D. The gDIS Algorithm

The pseudo-code of gDIS for estimating PDE, PIE and STE is presented in Algorithm 1.

## E. Datasets

Each unit $i$ has only one observed treatment $T_i$ and outcome $Y_i$ (the factual outcome), making direct causal effect estimation challenging due to unobservable counterfactuals. Since evaluating causal inference methods requires ground truth ITE (Individual Treatment Effect) values (which are impossible to obtain from observational data alone), following previous works (Jiang & Sun, 2022; Chen et al., 2024), we create semisynthetic datasets where the network structure (features and topology) is real, but treatments and potential outcomes are simulated.

The semisynthetic datasets are generated from two real-world social networks:

**BlogCatalog**[2] (**BC**): BlogCatalog is a social network where users publish blogs. In this dataset, each unit represents a blogger, and each edge represents a friendship link between bloggers. The features are represented as keywords extracted from the bloggers' posts. This dataset provides a rich network structure, with nodes connected by social relationships that capture the peer exposure.

**Flickr**[3]: Flickr is a social network for sharing images and videos. Here, each unit is a user, and each edge represents a social connection between users. Each user's features are represented by a list of tags that indicate their interests, forming a high-dimensional feature space. More details on these datasets are summarized in Table 4.

*Table 4.* Detailed information of datasets. A unit (node) represents a user, and an edge represents a social connection between users.

|  | BlogCatalog | Flickr |
| --- | --- | --- |
| # of Users | 5,196 | 7,575 |
| # of Features | 8,189 | 12,047 |
| # of Links | 171,743 | 239,738 |

---

[2] https://www.blogcatalog.com/
[3] https://www.flickr.com/

---

**Algorithm 1** gDIS (**g**roup-level P**D**E, P**I**E, and **S**TE)

---

1: **Input:** $\mathbf{X}, \mathbf{T}, \mathbf{W}_t, \mathbf{W}_y, \mathbf{W}_x, \mathbf{Y}$; learning rate $lr$; HSIC regularization parameter $\lambda$; adjacency matrix of data $\mathbf{A}$; $s$: number of epochs for Stage 1, $o$: number of epochs for Stage 2; $m$: number of samples.
2: **Output:** Estimated PDE, PIE, and STE.
3: **Initialize:** $f_1 \leftarrow \text{GNN}_2\text{-Attn-FC}_1, \quad f_2 \leftarrow \text{GNN}_2\text{-Attn-FC}_3$.
4: ▷ $f_1$ is for estimating $P(W_{y_i} \mid W_{t_i}, W_{x_i})$.
5: ▷ $f_2$ is for estimating $P(Y_i \mid W_{y_i}, W_{t_i}, W_{x_i})$.
6: **for** $e = 1, \ldots, s$ **do**
7:     ▷ Stage 1: estimate parameters for $f_1$
8:     $\hat{W}_{y_i} \leftarrow f_1(W_{t_i}, W_{x_i})$
9:     Compute loss $\mathcal{L}_1$ via Eq. (11).
10:     Update $f_1$ by descending $\nabla_{f_1}\mathcal{L}_1$.
11: **end for**
12: **for** $e = 1, \ldots, o$ **do**
13:     ▷ Stage 2: estimate parameters for $f_2$
14:     $\hat{Y}_i \leftarrow f_2(\hat{W}_{y_i}, W_{t_i}, W_{x_i})$
15:     Compute loss $\mathcal{L}_2$ via Eq. (12).
16:     Update $f_2$ by descending $\nabla_{f_2}\mathcal{L}_2$.
17: **end for**
18: **for** $i = 1, \ldots, m$ **do**
19:     ▷ Stage 3: density estimation
20:     Estimate densities $P(\mathbf{X}_i)$ and $P(\mathbf{W}_{x_i})$ via Eq. (15)–(16).
21: **end for**
22: Compute PDE, PIE and STE via Eq. (5)–(7) respectively.
23: **Return:** PDE, PIE, and STE.

---

Given the high-dimensional and sparse nature of the original features, we follow the approach in (Jiang & Sun, 2022; Chen et al., 2024) and apply Latent Dirichlet Allocation (LDA) (Blei et al., 2003) to reduce the feature dimension $k$ to 10. To partition the network, we utilize the **METIS** (Karypis & Kumar, 1998) graph partitioning tool, which efficiently handles large-scale graphs. Further details and case studies regarding METIS can be found in the official documentation at `https://metis.readthedocs.io/en/latest/index.html#module-metis`. To reduce memory consumption and streamline the analysis, we construct a new graph by removing isolated nodes that do not contribute to the network structure. The resulting partitioned data distribution is shown in Table 5.

*Table 5.* Data distribution after graph partitioning. Each tuple $(m, k)$ indicates the number of nodes $(m)$ and the feature dimension $(k)$.

| Dataset | Train | Validation | Test |
|---|---|---|---|
| BlogCatalog | (1722, 10) | (1733, 10) | (1731, 10) |
| Flickr | (1557, 10) | (2526, 10) | (1829, 10) |

## F. Simulation

Treatments and potential outcomes are simulated according to the underlying causal DAG, as shown in Fig. 2(a), allowing for the ground truths of causal effects in terms of evaluating our causal inference method.

**Treatments Simulation.** The features act as pre-treatment variables (i.e., features are parent nodes of the treatment) and influence the treatment. When simulating treatment generation, to account for the varying interference between node features, we use cosine similarity divergence to measure the similarity between connected units.

We calculate each individual's probability $p(T_i \mid \mathbf{X}_i, \{\mathbf{X}_j\}_{j \in \mathcal{N}(i)})$ of receiving treatment. $\mathbf{X}_i$ and $\{\mathbf{X}_j\}_{j \in \mathcal{N}(i)}$ denote the feature vector of individual $i$ and the feature vector of their neighbor $j$, respectively.

---

**Algorithm 2** Gibbs Sampling Algorithm

---
 1: **Input:** Initial values $Wy_0 = 0$, $Y_0 = 0$; number of iterations $m$.
 2: **Output:** $Wy^m$, $Y^m$.
 3: **for** $g = 1$ **to** $m$ **do**
 4:     **for** $i = 1$ **to** $n$ **do**
 5:         Compute $Wy_i^{g-1} = \sum_{j \in \mathcal{N}_i} w_{ij} Y_j^{g-1}$
 6:         Compute $Y_i^g = f(Wy_i^{g-1}, \mathbf{X}_i, \mathbf{X}_j, T_i, T_j)$
 7:     **end for**
 8:     Check for convergence using $\mathbf{Y}$ vectors.
 9:     **if** convergence is achieved **then**
10:         **break**
11:     **end if**
12: **end for**

---

$$p(T_i = 1 \mid \mathbf{X}_i, \{\mathbf{X}_j\}_{j \in \mathcal{N}_i}) = \sigma(\sigma(\alpha_0 \mathbf{w}_0 \mathbf{X}_i) + \sigma(\alpha_1 \sum_{j \in \mathcal{N}_i} w_{ij} \mathbf{w}_1 \mathbf{X}_j)) \tag{17}$$

where $\mathbf{w}_0$ and $\mathbf{w}_1$ are feature weight vectors drawn from the uniform distribution $\mathcal{U}(-1, 1)$, $w_{ij}$ is the weight indicating the influence of $j$ on $i$, and $\sigma$ is the sigmoid function. We define $\alpha_0 = 1$ as the fixed weight for individual features, and $\alpha_1 = 0.5$ as the fixed weight for neighbor features.

The treatment data is generated by the Bernoulli distribution (Chen & Liu, 1997):

$$T_i \sim \text{Bernoulli}\left(p(T_i = 1 \mid \mathbf{X}_i, \{\mathbf{X}_j\}_{j \in \mathcal{N}_i})\right) \tag{18}$$

**Potential Outcomes Simulation.** The potential outcome $Y_i$ is influenced by individual features $\mathbf{X}_i$, treatment status $T_i$, neighbor exposure $\{T_j\}_{j \in \mathcal{N}_i}$, neighbor features $\{\mathbf{X}_j\}_{j \in \mathcal{N}(i)}$, and neighbor outcomes $\{Y_j\}_{j \in \mathcal{N}_i}$. Following (Tchetgen Tchetgen et al., 2021; Zhao et al., 2024), we employ Gibbs sampling to iteratively generate $\{Y_j\}_{j \in \mathcal{N}_i}$ and $Y_i$ as follows:

$$\begin{aligned} p(Y_i \mid \mathbf{X}_i, T_i, \{\mathbf{X}_j\}_{j \in \mathcal{N}_i}, \{T_j\}_{j \in \mathcal{N}_i}, \{Y_j\}_{j \in \mathcal{N}_i}) = \sigma(\beta_0 \mathbf{w}_2 \mathbf{X}_i) + \sigma(\beta_1 \sum_{j \in \mathcal{N}_i} \mathbf{w}_3 w_{ij} \mathbf{X}_j) \\ + \beta_2 T_i + \beta_3 \sum_{j \in \mathcal{N}_i} w_{ij} T_j + \beta_4 \sum_{j \in \mathcal{N}_i} w_{ij} Y_j + \epsilon_i \end{aligned} \tag{19}$$

where $\epsilon_i$ is the noise term, and $\beta_0$, $\beta_1$, $\beta_2$, $\beta_3$, and $\beta_4$ are values randomly sampled from a uniform distribution within the interval $[0, 1)$, denoted $\beta_c \sim \mathcal{U}(0, 1)$ for $c = 0, 1, 2, 3, 4$. $\mathbf{w}_2$ and $\mathbf{w}_3$ are feature weight vectors drawn from the uniform distribution $\mathcal{U}(-1, 1)$. The iteration continues until the difference between $Y$ in successive steps is less than $1 \times 10^{-5}$, indicating that the generated data have reached a stable state. The Gibbs sampling algorithm is shown in Algorithm 2. Specifically, we first simulate the treatment of each node $T_i$ based on its own features $\mathbf{X}_i$ and its neighbors' features $\{\mathbf{X}_j\}_{j \in \mathcal{N}_i}$ via Eq. (17)–(18). Then, for every possible neighbor treatment configuration $w_{t_i}$ and its corresponding neighbor outcome configuration $w_{y_i}$, we apply Gibbs sampling according to Eq. (19) to generate the potential outcome $Y_i(t_i, w_{t_i}, w_{y_i})$, thus obtaining $Y_i$ under all scenarios $(t_i, w_{t_i}, w_{y_i})$. Finally, we plug these values into Definitions 3.4 and 3.5 to obtain the ground-truth PDE and PIE, and into Definition 3.7 to obtain the ground-truth STE.

## G. Experimental Setup and Configuration

In this section, we provide additional details about the experimental setup and configurations. Our proposed model is implemented using the Python libraries **TensorFlow** (Abadi et al., 2016) and **NetworkX** (Hagberg et al., 2008). We performed a grid search on the validation set to select the key parameters (learning rate, hidden dimensions, and regularization strength) that yielded the best performance, as shown in Table 6. Table 7 summarizes the parameter settings used for peer effect analysis on BlogCatalog and Flickr. Key parameter descriptions are as follows:

- Reps: The number of repetitions for each experimental group to ensure reliable results.

- Epoch: An epoch represents one complete forward and backward pass through the neural network.

- GNN Layers: The number of layers in the GNN.

- FC layers: The number of fully connected layers in the model.

- Hidden Dimension: The dimensionality of the hidden layers.

- $lr$: The learning rate.

- $\lambda$: The regularization parameter for the HSIC.

- $h$: The bandwidth parameter used in kernel density estimation.

*Table 6.* The parameter settings for the three stages across datasets.

| Stage | Parameter | BlogCatalog (BC) | Flickr |
|---|---|---|---|
| | Reps | 5 | 5 |
| | Epoch | 200 | 200 |
| Stage 1 | GNN Layers | 2 | 2 |
| | FC Layers | 1 | 1 |
| | Hidden Dimension | 8 | 8 |
| | $lr$ | 0.001 | 0.001 |
| | Reps | 5 | 5 |
| | Epoch | 150 | 180 |
| Stage 2 | GNN Layers | 2 | 2 |
| | FC Layers | 3 | 3 |
| | Hidden Dimension | 32 | 32 |
| | $lr$ | 0.01 | 0.001 |
| | $\lambda$ | $1 \times 0.3$ | $1 \times 0.3$ |
| Stage 3 | $h$ | 0.5 | 0.5 |

## H. Time Complexity

We analyze the time complexity of gDIS under different GNN structures and attention layers, as shown in Table 8. We observed that the runtime of the GAT layer (i.e., the graph-attention and GNN components) is significantly higher than that of the fully connected layer, indicating that the graph attention mechanism incurs substantial computational overhead for feature aggregation. Sparse graphs like Flickr exhibit lower runtime in the second stage despite their larger size, benefiting from a reduced average degree. Furthermore, **PyTorch Geometric** optimizes sparse graph processing, enabling efficient computation even for large-scale datasets.

*Table 7.* Hyperparameter search space for our method (gDIS).

| Stage | Parameter | Search Space |
|---|---|---|
| First Stage | $lr$ | {1e-4, 5e-4, 1e-3, 5e-3, 1e-2} |
| | GNN Layers | {1, 2, 3} |
| | FC Layers | {1, 2} |
| | Hidden Dimension | {8, 16, 32} |
| | Epochs | {100, 150, 180, 200} |
| Second Stage | $lr$ | {1e-4, 5e-4, 1e-3, 5e-3, 1e-2} |
| | GNN Layers | {1, 2} |
| | FC Layers | {1, 2, 3, 4} |
| | Hidden Dimension | {8, 16, 32, 64} |
| | Epochs | {100, 150, 180, 200} |
| Third Stage | KDE Bandwidth ($h$) | {0.3, 0.5, 0.7} |

*Table 8.* Layer Runtime on BC and Flickr Datasets: Training vs. Validation (seconds)

| Dataset & Stage | Layer | Runtime (s) | |
|---|---|---|---|
| | | Training | Validation |
| BC - First Stage | GAT Layer 1 | 0.00099 | 0.00100 |
| | GAT Layer 2 | 0.00100 | 0.00199 |
| | Fully Connected Layer | 0.00007 | 0.00007 |
| BC - Second Stage | GAT Layer 1 | 0.00300 | 0.00700 |
| | GAT Layer 2 | 0.00400 | 0.00701 |
| Flickr - First Stage | GAT Layer 1 | 0.00200 | 0.00100 |
| | GAT Layer 2 | 0.00400 | 0.00400 |
| | Fully Connected Layer | 0.00007 | 0.00007 |
| Flickr - Second Stage | GAT Layer 1 | 0.00200 | 0.00200 |
| | GAT Layer 2 | 0.00100 | 0.00200 |

Note: The timing measurements are taken for individual layers separately rather than for the entire model forward pass.

