# OpenReview forum: "Telling Peer Direct Effects from Indirect Effects in Observational Network Data"
_ICML.cc/2025/Conference — ICML 2025 poster_

### Official Review · Reviewer_oyya · 2025-02-14

**Overall Recommendation:** 3

**Summary:**

Estimating causal effects in observational network data is challenging due to peer interactions. Existing methods struggle to distinguish different types of peer effects. To address this, the proposed approach defines a general setting that considers peer direct effects, peer indirect effects, and individual treatment effects, along with their identification conditions. Using causal mediation analysis tailored for network data, the method differentiates these effects. It incorporates attention mechanisms to capture varying neighbor influences and employs multi-layer graph neural networks (GNNs) to explore high-order neighbor effects. Additionally, the Hilbert-Schmidt Independence Criterion (HSIC) enhances model robustness. Extensive experiments on semi-synthetic and real-world recommendation datasets validate the approach, with potential applications in social networks and public health interventions.

**Claims And Evidence:**

This paper is well-structured and supported by evidence.

**Essential References Not Discussed:**

No.

**Experimental Designs Or Analyses:**

Yes.

**Methods And Evaluation Criteria:**

Yes.

**Other Comments Or Suggestions:**

In simulation, more details regarding the baseline models are required. Specifically, what types of peer and self-treatment effects do they evaluate?
Typos: Line 041 (right)

**Other Strengths And Weaknesses:**

Strength.
They introduced the concepts of self-treatment effects, peer effects, and both direct and indirect peer effects within a many-to-one framework. Additionally, they proposed a novel gDIS algorithm, which demonstrates strong performance.

Weakness.
See my questions.

**Questions For Authors:**

1. Do you plan to move the motivating example of the product promotion campaign to the main body of the paper?
2. The notation W_{t_i} and W_{y_i} appears to be ambiguous. Do they depend on the realized values of  T  and  Y  for individual  i ?
3. You have defined peer effects through Definitions 3.5 and 3.6. However, I believe that Definition 3.6 is not a definition but rather a proposition.
4. Does your definition of peer effects align with previous works in the case of one-to-one relationships?
5. What does the total of PIE, PDE, and STE represent? Does it correspond the totoal effect of w_{t_i}' to w_{t_i} and 0 to 1 for individual i on outcome?
6. In simulations, why do the estimates for baselines and gDISs differ significantly? Do baseline estimates different peer and self-treatment effects? Is it possible to compute the ground truth values of PDE, PIE, and STE?
7. How did you compute error for each estimator? The values of counterfactuals in each effect are unobserved.

**Relation To Broader Scientific Literature:**

This contribution enhances our ability to optimize intervention strategies in public health, marketing, and social influence analysis. Peer effects is an emerging topic in causal inference.

**Theoretical Claims:**

Yes. The identifiability of PDE, PIE, and STE is a natural extension of previous works.

---

> ### Author Rebuttal · Authors · 2025-04-01
>
> We sincerely appreciate the reviewer's insightful comments and the high recognition of the value and importance of our work.
>
> **Comment:** *Baseline models details needed: Which peer and self-treatment effects do they evaluate?*
>
> In our simulation, the baseline models provide estimates of the overall peer effects (PE) and self-treatment effects (STE). Specifically, the baseline methods estimate:
>
> - **Peer Effects (PE):** This represents the aggregated influence of peers on an individual's outcome. These models capture the total effect that the treatments and outcomes of an individual's neighbors have on that individual but do not distinguish between the direct influence (Peer Direct Effects, PDE) and the indirect influence (Peer Indirect Effects, PIE).
>
> - **Self-Treatment Effects (STE):** This measures the effect of an individual's own treatment on their outcome.
>
> Thus, while the baseline models evaluate the combined peer effect and the self-treatment effect, they do not explicitly decompose the peer effects into PDE and PIE. Our proposed gDIS framework, on the other hand, is designed to disentangle these two components, offering a more detailed analysis of the peer influences within network data.
>
>
> **Q1:** *Product promotion example to main paper?*
>
> We plan to include the product promotion campaign example in the introduction, alongside the existing example from epidemiology. We will move the proof to the appendix to make space for the example.
>
> **Q2:** *$W_{t_i}$, $W_{y_i}$ notation appears to be ambiguous. Do they depend on the realized values of $T$ and $Y$ for individual $i$?*
>
> $W_{t_i}$ and $W_{y_i}$ do not depend on the realized values of $T_i$ or $Y_i$. They represent summary of the treatments and outcomes of unit $i$'s neighbors. $W_{t_i}$ and $W_{y_i}$ are illustrated in Figure 2b and explained in Line 313 (left column), and Appendix B.
>
> **Q3:** *Definition 3.6 as proposition?*
>
> We agree that Definition 3.6 may be more appropriately stated as a proposition as it presents a reformulation and decomposition of the peer effects introduced in Definitions 3.4 and 3.5.
>
> Our intention was to highlight this decomposition as a central conceptual component of our framework, which is why we initially presented it as a "definition." However, we appreciate the reviewer's point and will revise the manuscript to present it as a proposition.
>
> **Proposition** (Peer Effect Decomposition).
> The peer effect (PE) can be decomposed into the sum of the PDE and the PIE. That is,
>
>
> \\begin{aligned}
> \\text{PE}(w' _{t _i}) &= \\text{PDE}(w' _{t _i}) + \\text{PIE}(w' _{t _i}),
> \\end{aligned}
>
> \\begin{aligned}
> \\text{PE}(w _{t _i})  &= \\text{PDE}(w _{t _i}) + \\text{PIE}(w _{t _i}).
> \\end{aligned}
>
>
> The peer effect (PE) can be decomposed into the sum of the peer direct effect (PDE) and the peer indirect effect (PIE). That is,
>
> This proposition shows that the total peer effect integrates both the direct and indirect pathways of peer influence. Equation (3a) and Equation (3b) correspond to the vaccination and product promotion examples, respectively.
>
> **Q4:** *Alignment with previous work?*
>
> Our definition of peer effects aligns with prior works in the one-to-one setting, except that our peer treatment and outcomes are summaries of the treatments and outcomes of the neighbors of $i$, respectively.
>
> **Q5:** *Total effect representation?*
>
> Yes, you are right when there is no interaction between peer treatment $W_{t_i}$ and individual treatment $T_i$. We have assumed this in our work. This is a very reasonable assumption, as individual $i$'s treatment and peer treatment do not affect each other directly, and the reason for an individual to take the same treatment as their peers is because the individual and their peers have similar features (characteristics).
>
> **Q6:** *Baseline vs. gDIS differences?*
>
> a) In Table 2 of the paper, we made a mistake in summarizing the results for gDIS. We have updated the table and present the revised version at the following link (due to space limitations): https://anonymous.4open.science/r/icmlSupp-4556/table3.png
>
> b) The baseline methods' estimates of PE (peer effects) and STE (self-treatment effects) are formally consistent with our model; however, they do not explicitly decompose the PE into PDE (peer direct effects) and peer indirect effects (PIE).
>
> c) Using the given structural causal model (Equation (19) in Appendix F), we can obtain the potential outcomes under different treatment conditions.
>
> **Q7:** *Error computation for counterfactuals?*
>
> The potential outcomes with respect to $T_i$ for each $i$ are known since the structural causal model is assumed to be known for the synthetic dataset. Hence, the ground truth values of STE, PE, PDE, and PIE can be derived. Subsequently, compute the estimation error using the PEHE (Precision in Estimation of Heterogeneous Effect) metric, by comparing the estimated effect with the ground truth causal effect, i.e., the difference in potential outcomes.

---

### Official Review · Reviewer_dnoa · 2025-03-13

**Overall Recommendation:** 3

**Summary:**

This paper studies the causal effect estimation problem without SUTVA assumption. Specifically, the authors identify the overlooked problem that existing methods cannot distinguish between peer (in)direct effects and self-treatment effects. The authors propose a method called gDIS to estimate these estimands in the network scenarios. The proposed method is based on the back-door criteria, containing three stages to estimate the required density functions. The experimental results show its effectiveness.

## update after rebuttal
All my concerns are well addressed. I will keep my positive score.

**Claims And Evidence:**

The claims in this paper are supported by clear proofs and experimental results.

However, I am a bit confused by the DAG presented in Figure 2b.

Q1: Since it is 'Many-to-One' interference, units will affect each other. For the summary causal graph of __all units__, is it reasonable to use the DAG tool to represent the causal relationship for their interference?

**Essential References Not Discussed:**

The key related works are cited in the paper.

**Experimental Designs Or Analyses:**

The experimental designs and analyses are sound overall. However, I still have the following questions:

Q2: There is still a lack of comparison with existing peer direct/indirect effect methods cited in the introduction. Is it because they are of the 'One-to-One' type and cannot be applied to this scenario?

Q3: The potential outcome simulation is through an iteration process. How do you obtain the ground true direct/indirect effects?

**Methods And Evaluation Criteria:**

The proposed gDIS method makes sense to the network effect estimation problem.

**Other Comments Or Suggestions:**

typos:

1. Line 042: ana -> and
2. Line 087: method which do not.
3. Line 109: methods not -> methods do not
4. Line 198: frrom -> from

**Other Strengths And Weaknesses:**

Strengths

1. The setting is interesting.
2. The proposed estimator is reasonable and effective.

Weaknesses

1. Lack of comparison with PIE/PDE estimators.

**Questions For Authors:**

Please see Q1-Q3 in the above discussion.

Q4: Could you provide more experimental results with a wider range of $\lambda$? The paper claims $\lambda=0.3$ is optimal, but it seems only suboptimal as the curves in Figure 5 are still descent as $\lambda$ increases.
Q5: Could you clarify how to choose the other hyperparameters shown in Table 6 and provide the hyperparameter space of the compared baselines?

**Relation To Broader Scientific Literature:**

The studied problem is an overlooked problem, and this paper provides a practical and reasonable solution to estimating PDE/PIE and STE.

**Theoretical Claims:**

I checked all proofs roughly and did not find any problems.

---

> ### Author Rebuttal · Authors · 2025-04-01
>
> We thank the reviewer for the valuable comments. Detailed responses to each specific comment are provided below.
>
> **Weakness 1:** *Lack of comparison with PIE/PDE estimators.*
>
> The PIE/PDE estimators cited in our introduction (e.g., VanderWeele et al., Shpitser et al.) are designed for One-to-One interference settings, assuming dyadic interactions (e.g., household or pairwise influence). Our work focuses on the more general and realistic Many-to-One network setting. As such, these estimators cannot be directly or fairly compared with our method. Instead, we compare with recent network-level causal inference methods (e.g., NetEst, TNet, 1-GNN), which are designed to handle complex interference and represent the state-of-the-art in our setting.
>
> ---
> **Other Comments Or Suggestions: Typos**
>
> We thank the reviewer for the careful reading. We have corrected all.
>
> ---
>
> **Question 1:** *Since it is "Many-to-One" interference, units will affect each other. For the summary causal graph of all units, is it reasonable to use the DAG tool to represent the causal relationship for their interference?*
>
> The DAG shown in Figure 2b illustrates the unit-level causal relationships from the perspective of a single unit $i$, depicting how $i$'s outcome is affected by their own treatment and neighbors’ treatments and outcomes. This type of localized DAG is commonly used in the literature on causal inference with interference [1].
>
> [1] Jiang, Song, and Yizhou Sun. "Estimating causal effects on networked observational data via representation learning." *Proc. ACM Int. Conf. Inf. & Knowl. Manag.* 2022.
>
> ---
>
> **Question 2:** *There is still a lack of comparison with existing peer direct/indirect effect methods cited in the introduction. Is it because they are of the "One-to-One" type and cannot be applied to this scenario?*
>
> Yes. The methods cited in the introduction (e.g., VanderWeele et al., Shpitser et al.) are designed for One-to-One settings and assume dyadic interactions. Our setup considers Many-to-One interference, which is more general and realistic in network data. These earlier methods cannot be directly or compared in our setting. Hence, we have compared with recent methods (e.g., NetEst, TNet, 1-GNN) that support many-to-one relationships.
>
> ---
>
> **Question 3:** *The potential outcome simulation is through an iteration process. How do you obtain the ground true direct/indirect effects?*
>
> As stated in [2-3], Gibbs sampling provides a practical method for approximating the causal effects of interest in the presence of complex interdependencies among individuals in a network. This is achieved by iteratively sampling from the conditional densities of each individual. Our iteration continues until the difference between the values of $Y$ in successive steps is less than $1 \times 10^{-5}$, thereby ensuring that the generated data has reached a stable state. Then, using a given structural causal model (Equation 19 in the Appendix F), we can obtain the potential outcomes under different treatment conditions $T$. The ground true direct/indirect effects can be obtained from the potential outcomes.
>
> [2] Zhao, Ziyu, et al. "Learning individual treatment effects under heterogeneous interference in networks." *ACM Trans. Knowl. Discov. Data* 18.8 (2024): 1-21.
> [3] Tchetgen Tchetgen, Eric J., Isabel R. Fulcher, and Ilya Shpitser. "Auto-g-computation of causal effects on a network." *J. Am. Stat. Assoc.* 116.534 (2021): 833-844.
>
> ---
> **Question 4:** *Could you provide more experimental results with a wider range of $\lambda$? The paper claims $\lambda = 0.3$ is optimal, but it seems only suboptimal as the curves in Figure 5 are still descending as $\lambda$ increases.*
>
> In our experiments, we evaluated $\lambda$ values ranging from 0 to 0.5 (results available at the [anonymous link](https://anonymous.4open.science/r/icmlSupp-4556/2.jpg)). We found that $\lambda = 0.3$ produced the best overall performance. When $\lambda$ exceeds 0.3, it starts to over-penalize the feature representations, leading to a decline in performance.
>
> ---
>
> **Question 5:** *Could you clarify how to choose the other hyperparameters shown in Table 6 and provide the hyperparameter space of the compared baselines?*
>
> For our method, we performed a grid search on the validation set to select the key parameters (learning rate, hidden dimensions, and regularization strength) that yielded the best performance (shown in Table 6, Appendix H). Following this comment, we have updated the search space for these key parameters in Appendix H, as shown in [anonymous link](https://anonymous.4open.science/r/icmlSupp-4556/table1.png). For benchmark methods, we followed the recommended settings (as shown in [anonymous link](https://anonymous.4open.science/r/icmlSupp-4556/table2.png)) from their original papers and publicly available implementations.

---

### Official Review · Reviewer_GreV · 2025-03-17

**Overall Recommendation:** 3

**Summary:**

This paper focuses on differentiating between various types of causal effects in network data: peer-direct effects (PDE), peer-indirect effects (PIE), and self-treatment effects (STE). The authors propose a general setting to identify and estimate these effects, with theoretical identification conditions and proofs. They developed a method called gDIS (group-level Direct and Indirect effects estimator), which leverages graph neural networks (GNNs) with attention mechanisms and Hilbert-Schmidt Independence Criterion (HSIC) regularization to estimate these effects.

**Claims And Evidence:**

The claims regarding the identification of causal effects appear well-supported by the theoretical results. The empirical claims about the superior performance of gDIS are backed by comprehensive experimental results comparing against several baselines.

**Essential References Not Discussed:**

No major omissions were found, though recent work on continuous treatment effects in networks could be relevant.

**Experimental Designs Or Analyses:**

The experimental design comparing against multiple baselines and analyzing performance under various conditions (e.g., treatment flip rates, hyperparameter sensitivity) is thorough. The simulation of treatments and outcomes follows established practices in the causal inference literature.

**Methods And Evaluation Criteria:**

The proposed methods appear appropriate for the problem. The evaluation of semi-synthetic datasets with ground truth is a standard approach in causal inference research, and the metrics used (MSE and PEHE) are appropriate for evaluating causal effect estimates.

**Other Comments Or Suggestions:**

None

**Other Strengths And Weaknesses:**

Strengths:
1. The paper addresses an important gap in the literature by differentiating between direct and indirect peer effects in network settings, which is crucial for many real-world applications such as public health interventions and marketing campaigns.
2. The theoretical foundation is solid, with clear identification conditions and proofs that provide guarantees for the proposed methods.
3. The use of causal mediation analysis principles to differentiate between different types of peer effects is elegant and well-executed.


Weaknesses:
1. The assumption of network unconfoundedness (Assumption 3.8) is quite strong and may be violated in many real-world settings. While the authors acknowledge this limitation in the conclusion, more discussion on potential approaches to relax this assumption would strengthen the paper.
2. The complexity of the model with multiple components (GNN, attention, HSIC) makes it potentially difficult to implement and tune for practitioners. A more detailed analysis of the relative contributions of each component could help guide practitioners on which aspects are most crucial.
3. The simulated outcomes using Gibbs sampling may not fully capture the complex interdependencies in real network data. A more detailed sensitivity analysis of different data generation processes would strengthen the validity of the results.
4. The paper focuses primarily on binary treatments, while in many real-world scenarios, treatments might be continuous or multi-valued. Extending the approach to handle such cases would increase its practical utility.
5. While the paper mentions the time complexity analysis in the appendix, a more thorough discussion of scalability to very large networks would be beneficial given the computational demands of GNNs with attention mechanisms.

**Questions For Authors:**

1. How robust is the approach to violations of the network unconfoundedness assumption? Have you conducted any sensitivity analyses to assess this?
2. Could the gDIS framework be extended to continuous treatments or more complex outcome structures (e.g., multivariate outcomes)?

**Relation To Broader Scientific Literature:**

The work clearly relates to the literature on causal inference in networks and mediation analysis, properly contextualizing its contributions in relation to existing approaches.

**Theoretical Claims:**

I reviewed the theoretical claims related to identification conditions in Section 4.1, including Lemma 4.1 and Theorem 4.2. The proofs appear sound, with appropriate application of do-calculus and backdoor adjustment principles.

---

> ### Author Rebuttal · Authors · 2025-04-01
>
> We thank the reviewer for their insightful comments and recognition of our work's importance.
>
> **W1:** Network unconfoundedness assumption... more discussion needed
>
> We plan to explore: a) Instrumental Variable methods [1] to introduce variables influencing treatment but not outcomes; b) Hidden confounder modeling [2] to leverage graph-based representation learning for capturing unobserved confounding; c) Causal discovery [3] to employ data-driven methods for identifying adjustment sets and uncovering hidden confounding pathways.
> 1. Angrist, J.D., et al. Identification of causal effects using instrumental variables.
> 2. Louizos, C., et al. Causal effect inference with deep latent-variable models.
> 3. Colombo, D., et al. Learning high-dimensional directed acyclic graphs with latent and selection variables.
> ---
> **W2:** Multiple components (GNN, attention, HSIC) - each component contributions
>
> We'll add to Section 4.2:
>
> >**GNN Layers:** GNNs capture network interactions. Without them, our model couldn't leverage network structure for modeling peer effects, leading to oversimplified estimations that ignore relational dependencies.
>
> >**Attention Mechanism:** Enables assigning different weights to neighbors based on feature similarity, useful where peer influence isn't uniform.
>
> > **HSIC Regularization:** Mitigates overfitting by encouraging independence between node features and embeddings, reducing spurious correlations.
> ---
> **W3:** More detailed sensitivity analysis of data generation processes needed
>
> Gibbs sampling provides a practical method for approximating causal effects with complex interdependencies in networks by iteratively sampling from conditional densities [4]. In our data generation process, we varied noise levels and found that—since our iteration continues until the difference between values of $Y$ in successive steps is less than $1 \times 10^{-5}$—the data exhibits high stability across different noise levels. This indicates our process effectively captures complex network interactions. We plan to explore additional data generation methods and conduct further sensitivity analyses.
>
> 4. Zhao, Z., et al. Learning individual treatment effects under heterogeneous interference in networks.
> ---
> **W4:** Extension to non-binary treatments would increase practical utility
>
> Our framework is flexible and does not rely on the treatment must be binary. When computing PDE and PIE, the neighbor treatment exposure variable $W_{t_i}$ supports continuous, multi-valued, or binary treatments. For STE, we've followed standard binary treatment approaches, but as a CATE (conditional average treatment effect) problem, it's extendable to continuous/multi-valued treatments [5-6].
>
> 5. Hirano, K., et al. The propensity score with continuous treatments.
> 6. Feng, P., et al. Generalized propensity score for estimating the average treatment effect of multiple treatments.
> ---
> **W5:** Scalability discussion needed on large-scale datasets
>
> We'll add the following to Appendix I:
>
> > Although attention mechanisms introduce additional overhead compared to standard GNNs, we optimize sparse graph processing using PyTorch Geometric [7], enabling efficient computation even on large-scale datasets. Additional strategies can enhance scalability: sampling-based techniques like those in GraphSAGE [8] can limit neighbors sampled for each node, reducing computational demands while preserving structural information. Approximation or sparsification techniques for the attention layer can also alleviate computational burdens [9].
>
> 7. Fey, M., et al. Fast graph representation learning with PyTorch Geometric.
> 8. Hamilton, W., et al. Inductive representation learning on large graphs.
> 9. Child, R., et al. Generating long sequences with sparse transformers.
> ---
> **Q1:** Robustness to violations of network unconfoundedness
>
> We're currently analyzing violations of the network unconfoundedness assumption. Our preliminary expectation is that robustness depends on factors like the strength of association between hidden confounders and treatment/outcome. Future work will include comprehensive sensitivity analyses to quantify how these factors affect performance.
>
> ---
> **Q2:** Extension to continuous treatments or complex outcomes
>
> Yes, gDIS can handle continuous treatments and complex outcome structures. Our framework is flexible with no binary treatment requirement. When computing PDE and PIE, the neighbor treatment exposure variable $W_{t_i}$ accommodates any treatment type. For STE, our implementation follows standard causal inference with binary treatment effects, but as STE estimation is a typical CATE problem, our approach can extend to continuous or multi-valued treatments using existing techniques [6].
> For multivariate outcomes, the framework can be adapted using joint modeling approaches (e.g., multi-output regression [10]) to capture dependencies among multiple outcomes.
>
> 10. Sener, O., et al. Multi-task learning as multi-objective optimization.

---

### Official Review · Reviewer_igcq · 2025-03-25

**Overall Recommendation:** 3

**Summary:**

The paper addresses the challenge of estimating treatment effects in observational network data with network interference. The authors propose a framework to decompose peer effects into direct and indirect peer effects and provide theoretical analyses of the identification conditions. Additionally, the paper introduces gDIS, a novel algorithm that leverages graph neural networks and attention mechanisms to estimate effects in network data.

**Claims And Evidence:**

Yes, the paper provides the theoretical proofs and empirical evidence to support the claims.

**Essential References Not Discussed:**

Essential references were discussed in this work.

**Experimental Designs Or Analyses:**

I did not find particular issues with the experimental design.

**Methods And Evaluation Criteria:**

The authors validated the proposed approach on simulated and semi-synthetic data and compared the model performance with six baselines.

**Other Comments Or Suggestions:**

No

**Other Strengths And Weaknesses:**

**Strengths:**
1. The paper tackles a significant challenge in estimating peer effects under network interference. The paper employs GNN with attention mechanisms to capture network structure.
2. Theoretical justification and empirical results of comparison with other classic estimators are provided.

**Weaknesses:**
1. The effectiveness of the proposed approach depends on strong assumptions, which may be challenging to satisfy in real-world observational data.

**Questions For Authors:**

1. Are there any specific requirements for the network structure in order to ensure the validity of the proposed approach?

**Relation To Broader Scientific Literature:**

This paper fits well into the classic literature on estimating peer effects on networks for observational studies.

**Theoretical Claims:**

Yes

---

> ### Author Rebuttal · Authors · 2025-04-01
>
> We thank the reviewer for the valuable comments. Detailed responses to each specific comment are provided below.
>
> **W1:** *The effectiveness of the proposed approach depends on strong assumptions, which may be challenging to satisfy in real-world observational data.*
>
> Our method is based on three key assumptions: Network Unconfoundedness, Network Consistency, and Network Overlap. These assumptions are commonly assumed in causal inference in networked data [1].
>
> In this paper, our focus is on the challenge of distinguishing and estimating different types of causal effects in network data (i.e., the *Peer Direct Effect (PDE)*, *Peer Indirect Effect (PIE)*, and *Self-Treatment Effect (STE)*), and we adopt the assumptions commonly used in the network causal inference literature.
>
> We have included discussions regarding the limitations of assumptions in the "Limitations & Future Work" section.
>
> - [1] Jiang, Song, and Yizhou Sun. "Estimating causal effects on networked observational data via representation learning." *Proceedings of the 31st ACM International Conference on Information & Knowledge Management*, 2022.
>
> **Q1:** *Are there any specific requirements for the network structure in order to ensure the validity of the proposed approach?*
>
> Our proposed approach does not impose structural requirements on the network (e.g., specific topology or connectivity patterns). However, we do assume the causal relationships are as in Figure 2(b), which are realistic.

---

### Decision · Program_Chairs · 2025-05-01

**Decision:**

Accept (poster)

**Comment:**

All reviewers are leaning weakly positive on this work and agree it addresses an interesting topic on the decomposition of peer effects in causal inference with network interference. Some concerns remain around the strong assumptions underlying the method.